# Main Diagnostic Pitfalls in Reading the Sacroiliac Joints on MRI

**DOI:** 10.3390/diagnostics11112001

**Published:** 2021-10-28

**Authors:** Sammy Badr, Thibaut Jacques, Guillaume Lefebvre, Youssef Boulil, Ralph Abou Diwan, Anne Cotten

**Affiliations:** 1Department of Musculoskeletal Radiology, Lille University Hospital, 59000 Lille, France; sammy.badr@chu-lille.fr (S.B.); thibaut.jacques@chu-lille.fr (T.J.); guillaume.lefebvre59@gmail.com (G.L.); youssef.boulil@hotmail.fr (Y.B.); ralphaboudiwan@hotmail.com (R.A.D.); 2MABLab-Marrow Adiposity and Bone Lab ULR4490, University of Lille, 59000 Lille, France; 3Lille University School of Medicine, 59000 Lille, France

**Keywords:** anatomy, magnetic resonance imaging, post-partum period, osteoporotic fractures, sacroiliac, joint, sacroiliitides, spondylarthritis

## Abstract

Magnetic resonance imaging of the sacroiliac joints is now frequently performed to help identify patients with early axial spondyloarthritis. However, differential diagnoses exist and should be recognized. The aim of this article is to review the most frequent differential diagnoses that may mimic inflammatory sacroiliitis in clinical practice.

## 1. Introduction

Magnetic resonance imaging (MRI) of the sacroiliac joints (SIJ) is now frequently performed to help identify patients with early axial spondyloarthritis (axSpa) before the development of the radiographic features of sacroiliitis. Active inflammation on MRI that is highly suggestive of sacroiliitis is now included in the imaging arm of the ASAS criteria along with at least one other feature of axSpa [1,2]. The lesion definitions indicating signs of activity (bone marrow edema (BME), capsulitis, joint space enhancement, inflammation at the site of erosion, enthesitis, joint space fluid) or structural changes (erosion, fat lesion, sclerosis, ankylosis, non-bridging bone bud) have been standardized by the ASA-MRI working group [3] (Figure 1). However, there is a significant rate of diagnostically relevant SIJ abnormalities on MRI in patients without axSpa [4,5]. Differential diagnoses exist, some of which are encountered more frequently than inflammatory sacroiliitis. Their recognition is helpful in avoiding false positive diagnosis of sacroiliitis. The aim of this article is to review the most frequent differential diagnoses that may mimic inflammatory sacroiliitis in clinical practice. 

## 2. Mechanical Changes and Osteoarthritis 

The first question concerning changes in subchondral bone marrow in the SIJ is where are they located? This element is crucial for differentiating between mechanical changes and sacroiliitis. Mechanical changes are the main pitfall in reading MR images of the SIJ. They are caused by mechanical overload and may lead to osteoarthritis. They are typically located in the anterior part of the mid-third of the SIJ, which is the area where mechanical stress is concentrated (Figure 2). Due to the C shape of the cartilaginous part of the SIJ, the mechanical area is located on the most anterior coronal oblique sections oriented on the S1–S3 axis [6,7] (Figure 2). Abnormal bone marrow signal intensities (edema, sclerosis, or both) seen on one, two, or more rarely three and exceptionally four contiguous slices in this sole specific location are irrelevant when limited in size, well delineated, and when they occur without significant erosion (Figure 3). These findings are in contrast with BME and/or sclerosis seen in any other parts of the joint and are not involved in mechanical loading, such as the proximal or distal third or the posterior part of the joint, where involvement is highly suggestive of sacroiliitis. Axial orthogonal oblique images easily confirm the anterior location of mechanical changes, which are often triangular in shape and bilateral. Minor osteophytosis may be associated. These MR changes are usually asymptomatic, but some may be associated with pain [6].

Interestingly, in young athletes, while the same area can be affected, BME may also be encountered in the distal SIJ, probably reflecting a different distribution of mechanical forces [8].

## 3. Osteitis Condensans Ilii and Post-Partum Changes

### 3.1. Osteitis Condensans Ilii

Osteitis condensans ilii is usually asymptomatic, found incidentally on imaging or, less commonly, associated with low back pain. Prevalence is higher in postpartum women, but it can also be seen in women without a history of pregnancy and in men. Extensive mechanical stress is usually suggested to be the cause, but the pathophysiology of this self-limiting condition is still not completely clear [9,10,11].

Osteitis condensans ilii has historically been characterized by predominant sclerosis of the ilium as seen on pelvis X-rays and CT. The sclerosis is typically dense, compact, homogeneous, well-delineated, triangular in shape, and usually bilateral. Although typically predominating in the anterior part of the mid-third of the SIJ, it is much more extensive than the mechanical changes described above. 

MRI shows the sclerotic areas as being hypointense on T1- and T2-weighted images, relatively homogeneous, well-delineated, and frequently extending beyond this area although these areas are predominantly located in the anterior part of the mid-third of the SIJ (Figure 4) [12]. These areas are sometimes surrounded by a mild rim of BME [11]. This extensive subchondral sclerosis contrasts with findings of no or very moderate joint erosion, BME, or fatty metaplasia, unlike in sacroiliitis [12]. Ankylosis is not observed in this condition. 

### 3.2. Post-Partum

Post-partum changes seen on MRI have been well described. Pregnancy-induced BME in the SIJ as a result of prolonged mechanical stress was reported in 60–64% of women during the early postpartum period [13,14,15]. Presence and extent do not correlate with symptoms of pain (Figure 5) [13]. Distribution and extent are indistinguishable from the BME seen in inflammatory sacroiliitis [13]. Subchondral sclerosis is also frequently observed, while erosion is rarely encountered, which is in contrast to inflammatory sacroiliitis (Figure 5) [10,13]. 

BME in the SIJ tends to disappear over time [16,17], decreasing to 15-17% at 6 months post-partum [15,18] to 12% at 12 months [15] and to 5% after 16 months [17]. In contrast, SIJ subchondral sclerosis can persist into the late postpartum period [16,19]. In one study, the number of children a woman had correlated positively with SIJ sclerosis, whereas the birth method had no impact [19].

## 4. Anatomical Variants

At least seven SIJ variants have been described in the literature [20,21] and can involve the cartilaginous or the ligamentous part of the joint. Interestingly, patients with such variants more frequently complain of low-back pain [22]. Only variants associated with misleading MR features that might mimic sacroiliitis will be presented here. 

### 4.1. Dysmorphic SIJ Aspect

A particular unilateral or bilateral dysmorphic aspect of the cartilaginous part of the SIJ was recently described on MR images [20]. It seems to be the most frequent anatomical variant of the SIJ and has a prevalence of 17–21% [20,21]. It is characterized by an elongated curved posterior joint with a convex sacral surface border protruding into the adjacent iliac bone. Continuity with the regular cartilaginous joint differentiates it from an accessory facet joint. This dysmorphic SIJ needs to be recognized, as it can be associated with sclerotic or fatty structural and/or edematous changes that might be interpreted as features of sacroiliitis, particularly on coronal oblique images (Figure 6). This is all the more important as these abnormal signal intensity changes are located in the distal third of the SIJ, a location which would otherwise suggest sacroiliitis, since it is not located in the mechanical area.

However, axial oblique images allow a clear depiction and analysis of this variant (Figure 5) [20]. They demonstrate that the bone marrow changes are focal, limited in size, and centered by the dysmorphic joint at the distal and posterior part of the cartilaginous joint. These features could be related to microtraumatic changes in this focal area. Prominent cystic changes can also be seen, but there is no significant erosion.

Of course, inflammatory sacroiliitis can also develop in this area, including in the presence of a dysmorphic joint. However, the MR features are different and include a more extensive BME, which is not centered exclusively on the dysmorphic joint.

### 4.2. Accessory SIJ 

The accessory SIJ is different from the dysmorphic joint described above, as it is located dorsally in the ligamentous part and at a distance from the regular cartilaginous joint. It is defined as an articulation with congruent, flat, and slightly convex or concave sacral and iliac facets [20,21]. It can be unilateral or bilateral, and it has a prevalence of 16–20% [20,21]. It can be associated with sclerotic, fatty, and/or edematous changes in the facing bones and is likely caused by microtrauma [20]. An association between these features and low-back pain is controversial and needs to be further assessed. These MR changes should not be misinterpreted as features of sacroiliitis, particularly on coronal oblique images. This variant is much more easily identified on axial oblique images (Figure 7) [20].

This variant should be differentiated from the iliosacral complex, which can be defined as a ≥ 5 mm focal osseous protrusion of the iliac bone in the ligamentous part of the joint without visible articulation [20,21]. The iliosacral complex is rarely associated with BME, which is most often limited in volume and is probably strain-related [21].

### 4.3. Isolated Synostosis

Primary unilateral SIJ synostosis is an exceptional variant [20,23]. It partially involves the SIJ (mid-third, at the first sacral foramen) and is well depicted on both the axial and coronal oblique images [20]. It may mimic acquired SI ankylosis resulting from ankylosing spondylitis or infectious sacroiliitis. However, the residual SIJ and contralateral joint are completely normal and do not show any inflammatory or structural changes [20,23]. 

## 5. Vessels

Longitudinal vessels are abundant in the transitional cartilaginous-ligamentous portions of the SIJ. The partial volume effect on these vessels on coronal oblique images may mimic edematous changes and consequently either enthesitis or BME [8]. However, the tubular appearance or cross-sectional rounded aspect of these vessels is usually well demonstrated on axial oblique images, which is in contrast to the ill-defined edema seen in the enthesitis of the ligamentous portion of the SIJ.

## 6. Pediatric SIJ 

MR assessment of pediatric SIJ may be challenging due to normal developmental changes that occur in the immature skeleton [24]. In children, cartilage epiphysis, underlying newly formed subchondral bone as well as red bone-marrow can all appear as areas of moderately increased signal on T2-weighted images, particularly on the sacral side of the joint. These normal MR features result in smooth symmetric linear bands with a high T2 signal along the craniocaudal length of both SIJs and parallel to them [25,26]. The expressions “metaphyseal-equivalent signal intensity” and “bone flaring” are frequently used in the literature to describe this appearance, which is influenced by age and gender [25,26]. Different types of high signal have been described in the sacrum [25,26]. Type 1, which is characterized by distinct homogeneous increased signal intensity areas that extend along unfused sacral apophyses, and type 2, which is more indistinct and only extends partially along incompletely fused sacral apophyses, are present in most prepubertal children (Figure 8). By the time these children approach skeletal maturity, type 1 is no longer present, and type 2 is only detectable in a minority of boys [25,26]. Iliac flaring is less frequent and is mainly seen in the distal part of the joint. It is usually less prominent in the distal part of the joint than it is on the sacral side. 

Consequently, the MR features that should raise the suspicion of sacroiliitis include asymmetrical appearance, focal involvement, intense high signals (similar to the signals of the presacral veins) on fat-suppressed T2-weighted images, large area (depth from the articular surface ≥5–10 mm), and intensity/width that is greater on the iliac side than on the sacral side [25,26].

The presence of fluid in the SIJ space is reported to be rare; whether it should be considered a normal finding is debated [27]. Differentiation from adjacent cartilage is influenced by the type of MR sequence that is used. In clinical practice, a significant amount of fluid or an asymmetrical appearance should raise suspicion of abnormality. In case of doubt, the enhancement of the joint after gadolinium injection may be a useful complement. 

Cortical bone irregularities, blurring, or undulations mimicking erosion are also common and are most frequently observed within the upper iliac quadrants [25,28]. These pseudo-erosions are particularly frequent in the peripubertal phase. However, unlike true erosions, they usually have smooth and regular margins without any cortical lining defects and are not associated with adjacent sclerosis or fat infiltration.

## 7. Bone Insufficiency Fractures

Sacral bone insufficiency fractures result from normal mechanical stress on an abnormal underlying bone. They are most frequently seen in osteoporosis settings and thus are mostly seen in elderly females presenting with low-back pain without any history of significant trauma, although younger patients can also be affected. These fractures are frequently bilateral. They are usually vertical, through the sacral alae and paralleling the SIJ, and often associated with a transverse component that then resembles the capital letter H, hence the designation “Honda sign”, “H sign”, or “H pattern”.

On MRI, attention is usually drawn to a more or less extensive BME involving the sacral side of each SIJ. Fractures can be identified as thin hypointense irregular lines paralleling the SIJ [29] (Figure 9). BME exclusively or predominantly located on the sacral side is possible in sacroiliitis. However, the bilateral and relatively symmetrical distribution of the BME, sometimes with the aforementioned H pattern, the presence of fracture lines, the absence of structural changes in the SIJ, and the patient’s age should raise suspicion of sacral insufficiency fractures.

## 8. Infectious Sacroiliitis

Infectious sacroiliitis is not uncommon. It is more frequently encountered in young subjects [6]. Unfortunately, diagnosis is often delayed due to the insidious clinical presentation. Moreover, laboratory features such as leukocytosis and C-reactive protein (CRP) elevation are variable and inconstant [30]. Diagnosis is, however, critical, as urgent treatment is required to avoid increased morbidity [31].

Changes can be seen on MRI within 3 days of symptom onset [32]. Sacroiliitis is typically unilateral, and the MR aspect may be similar to that of inflammatory sacroiliitis, although it is usually more extensive [6]. MR features include extensive BME with frequent sacral predominance or distribution (in contrast to iliac predominance in axSpa) [9], abundant joint fluid, thick capsulitis, peri- and extra-articular edema, regional muscle swelling and edema, and fluid collections in the surrounding soft tissues (abscesses) (Figure 5 and Figure 10) [9,31,33]. Intravenous gadolinium administration is mandatory for the assessement of this disease. In the advanced stages, erosions, bony bridges, fatty replacement, and ankylosis may also be found [6]. 

## 9. Hyperparathyroidism

Extensive subchondral bone resorption that is predominantly or exclusively located on the iliac side of the SIJ can be observed in primary or secondary hyperparathyroidism, mimicking sacroiliitis (Figure 11). However, the huge widening of the joint seen in this condition is unusual in inflammatory sacroiliitis. Adjacent BME and sclerosis can be associated with bone resorption. Brown tumors can be seen in association with this diagnosis, particularly in patients with chronic kidney disease.

## 10. Malignancies

Finally, malignancies such as metastases, myeloma, or lymphoma may rarely be misleading when bone marrow replacement and/or osteolysis involve(s) the subchondral bone of the SIJ. However, the size of the lesion(s) in contrast with the absence of other typical features of sacroiliitis, and sometimes the presence of an adjacent soft tissue mass, usually help in the recognition of these disorders.

## 11. Conclusions and Perspectives

In conclusion, although the MR features of sacroiliitis in axSpa have been described in detail, several differential diagnoses have to be kept in mind. Many are best depicted on axial oblique images, which should be used in addition to coronal oblique T1 and fluid sensitive T2-weighted fat-saturated sequences if doubt is present. Diffusion-weighted imaging is a sensitive MRI sequence in the detection and quantification of active sacroiliitis but is not included in routine protocols [34]. 

New MR techniques, such as quantitative MRI [35,36], MRI-based synthetic CT [37], and machine-learning-supported texture analysis [38], could improve our understanding of SIJ changes on MRI in the future. However, whatever the MR protocol is, correlation with clinical data remains essential.

## Figures and Tables

**Figure 1 diagnostics-11-02001-f001:**
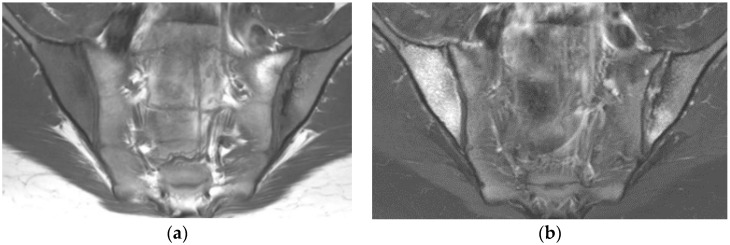
Sacroiliitis with bilateral BME predominating on the right side and erosions, sclerosis, and fat lesion on the left side (coronal oblique T1- (**a**) and fat-suppressed T2- (**b**) weighted images).

**Figure 2 diagnostics-11-02001-f002:**
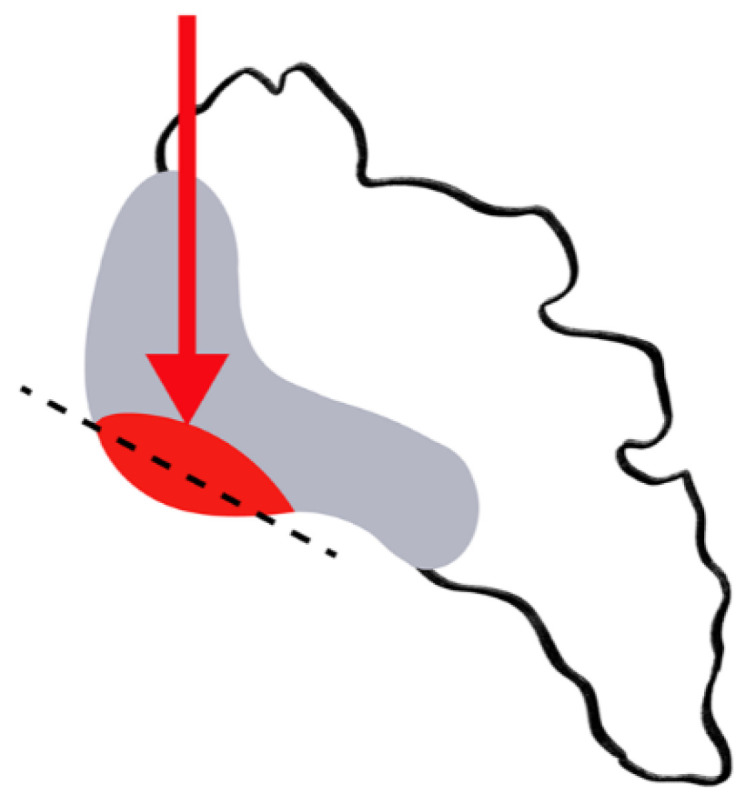
Drawing of the C-shaped cartilaginous part of the SIJ (grey color). Mechanical overload (red arrow) is concentrated in the anterior part of the mid-third of the SI (read area). This area is assessed on the most anterior coronal oblique sections (dotted line).

**Figure 3 diagnostics-11-02001-f003:**
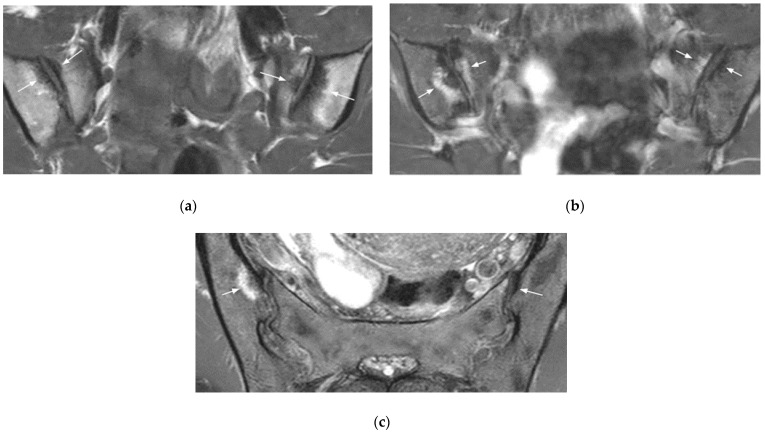
Bilateral mechanical changes in the SIJ (arrows) (coronal oblique T1- (**a**) and fat-suppressed T2- (**b**) weighted images, axial oblique T2-weighted image (**c**). The changes are mainly edematous on the sacral side of the SIJ, sclerotic on the right iliac side, and both edematous and sclerotic on the left iliac side. They are located at the anterior part of the mid-third of the SI.

**Figure 4 diagnostics-11-02001-f004:**
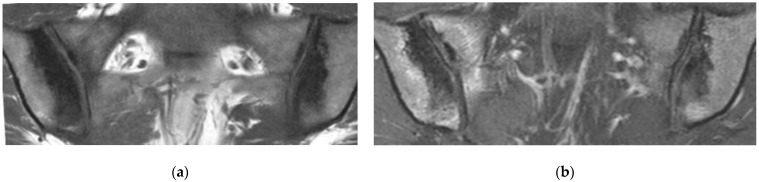
Osteitis condensans ilii in a 36-year-old woman (HLA-B27 negative) (coronal oblique T1- (**a**) and fat-suppressed T2- (**b**) weighted images). Extensive subchondral sclerosis at the iliac side is hypointense on T1- and T2-weighted images, well-delineated, and surrounded by a mild rim of BME, particularly on the right side.

**Figure 5 diagnostics-11-02001-f005:**
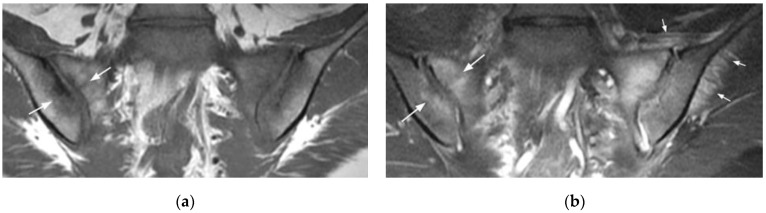
Post-partum changes (right SIJ) and infectious sacroiliitis by staphylococcus aureus (left SIJ) (coronal oblique T1- (**a**) and fat-suppressed T2- (**b**) weighted images). Right post-partum changes include mild sclerosis surrounded by BME (long arrows). On the left side, an extensive BME, hypointense on T1-, and hyperintense on T2-weighted images is associated with abundant joint fluid and extra-articular edema (small arrows).

**Figure 6 diagnostics-11-02001-f006:**
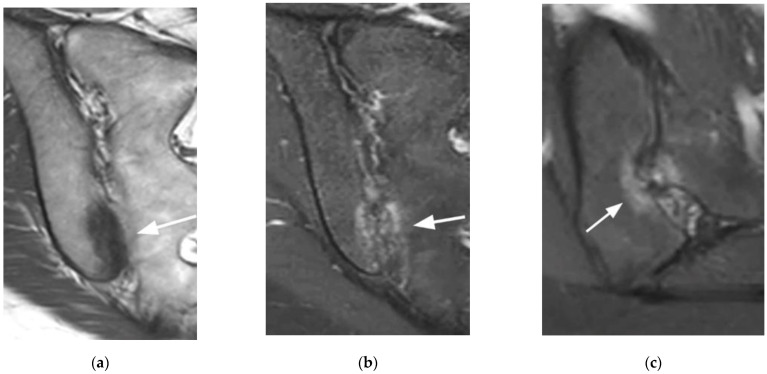
Dysmorphic SIJ (coronal oblique T1- (**a**) and fat-suppressed T2- (**b**) weighted images, axial oblique fat-suppressed T2- (**c**) weighted image). Edematous and sclerotic changes of the distal third of the SIJ mimicking sacroiliitis on the coronal images (arrow). On the axial oblique image, these features are focal, limited in size, and centered by the dysmorphic joint in the posterior part of the cartilaginous joint (arrow).

**Figure 7 diagnostics-11-02001-f007:**
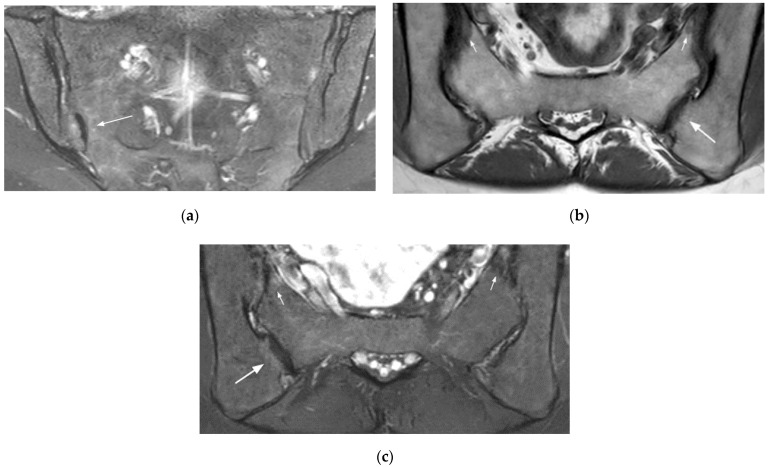
Right accessory SIJ (coronal oblique fat-suppressed T2- (**a**) weighted image, axial oblique T1- (**b**), and fat-suppressed T2- (**c**) weighted images). This unilateral accessory joint is mainly associated with sclerotic changes and is surrounded by a mild BME (long arrow). Note the associated mechanical changes in the SIJ (small arrows).

**Figure 8 diagnostics-11-02001-f008:**
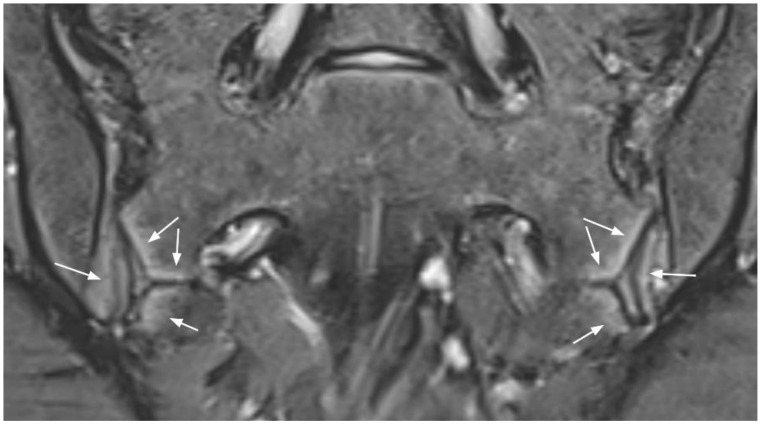
Normal type 2 high signal in a 15-year-old girl (coronal oblique fat-suppressed T2-weighted image). Linear band of high signal along the distal third of the SIJ that extends along partially fused sacral apophyses (arrows).

**Figure 9 diagnostics-11-02001-f009:**
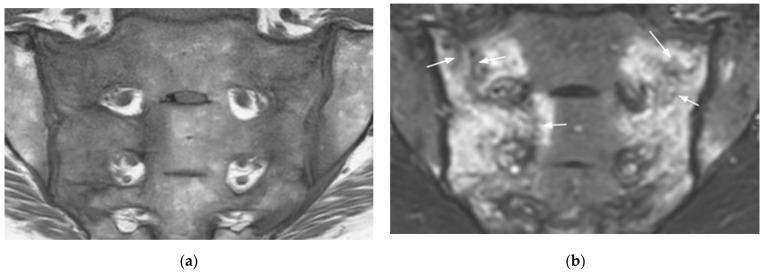
Sacral bone insufficiency fractures (coronal oblique T1- (**a**) and fat suppressed T2- (**b**) weighted images). Extensive BME on the sacral side of each SIJ containing thin irregular hypointense lines (fractures) (arrows).

**Figure 10 diagnostics-11-02001-f010:**
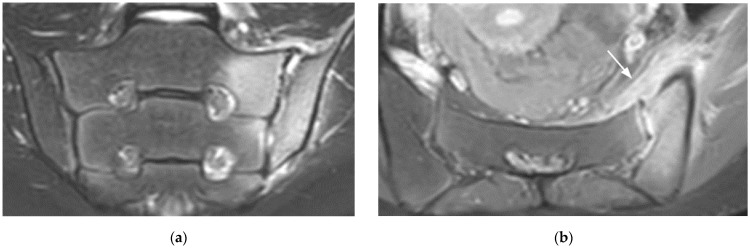
Left infectious sacroiliitis in a 14-year-old girl (coronal oblique T2- (**a**) and axial fat-suppressed post-gadolinium T1- (**b**) weighted images). Extensive BME, abundant joint fluid, joint space enhancement, and extra-articular inflammatory changes in the adjacent soft tissues and muscles (arrows).

**Figure 11 diagnostics-11-02001-f011:**
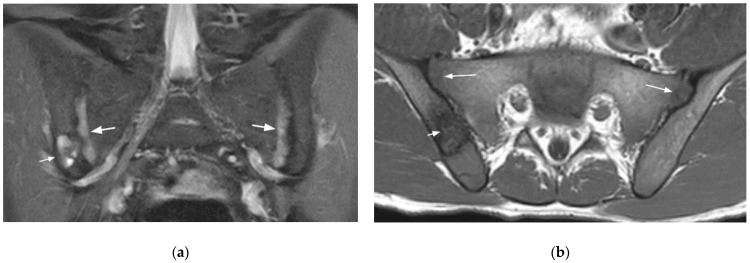
Subchondral bone resorption in a patient with chronic kidney disease (axial T1- (**a**) and coronal T2- (**b**) weighted images). Widening of the joint predominating on the iliac side (long arrows) associated with a brown tumor (small arrow).

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
