# Peer review of "Main Diagnostic Pitfalls in Reading the Sacroiliac Joints on MRI"

_diagnostics, 2021, doi:10.3390/diagnostics11112001_

Round 1

Reviewer 1 Report

  1. Introduction

Leave out CT features (line 18)

Replace SpA by axSpa

Insert ASAS criteria for diagnosis and elaborate on this; this is still in the research setting. Also the diagnosis AxSpa is not based on a MRI alone.

 (Ann Rheum Dis 2016 Nov;75(11):1958-1963; Ann Rheum Dis. 2019 Nov;78(11):1550-1558.)

The sentence (line 23-25) is too strong in my opinion. Knowledge of.. is necessary to avoid misinterpretation in early and more advanced AxSpA. What potential dangerous adverse effects?

  1. On conventional images it is the distal part, please insert

Fig.1. age and gender of the patient. HLA B27 -? Please check the description (left -right/ sacral-iliac side)

  1. Line 70 : see above: On conventional images it Is the distal part, please insert

Fig 3. Add: At the iliac side

Fig 4: proven infection ? images with contrast performed, this is obligatory? DD enthesitis (see 5)

  1. Can you give the prevalence of variants?
  2. Start with the vessels and their location . Subchondral edema can be mimicked. Elaborate how enthesitis looks like.
  3. –line 218 :add patient’s history (e.g. Radiation therapy)
  4. For this diagnosis MR with contrast is mandatory- please add this.

Line 233 replace:  dramatic by extensive

Fig 9. Please add age and gender of the patient . There is no fluid , but enhancing synovium seen , synovitis– add : no abces

  1. The screening for AxSpa is done with only coronal oblique images. So the conclusion would be that in case of doubt axial images can be of help. (Line 259 ). Please add that the clinical correlation is essential.

Author Response

  1. Introduction

- Leave out CT features (line 18). This has been deleted.

- Replace SpA by axSpa. This has been replaced.

- Insert ASAS criteria for diagnosis and elaborate on this; this is still in the research setting. Also the diagnosis AxSpa is not based on a MRI alone.  (Ann Rheum Dis 2016 Nov;75(11):1958-1963; Ann Rheum Dis. 2019 Nov;78(11):1550-1558.). It is now indicated that « active inflammation on MRI highly suggestive of sacroiliitis is now included in the imaging arm of the ASAS criteria along with at least one other feature of axSpa [1, 2] ».

-The sentence (line 23-25) is too strong in my opinion. Knowledge of.. is necessary to avoid misinterpretation in early and more advanced AxSpA. What potential dangerous adverse effects? The sentence has been corrected and the final part of the sentence (potential dangerous effect) has been deleted.

  1. - On conventional images it is the distal part, please insert. Mechanical overload is not concentrated in the distal part of the joint but at the anterior part of the middle third of the joint, whatever the imaging modality (MR, CT or X-rays). The distal part of the joint is not involved by mechanical changes (in contrast to sacroiliitis), unless there is a dysmorphic joint. The text has not been corrected.

- Fig.2 (now 4). age and gender of the patient. HLA B27 -? This information has been added.

- Please check the description (left -right/ sacral-iliac side). There was a mistake in the right-left side which has been corrected. Thank you.

  1. - Line 70 : see above: On conventional images it Is the distal part, please insert. Please look at answer 2.

- Fig 3(now 4). Add: At the iliac side. This has been corrected 

- Fig 4 (now 5): proven infection ? Infection by staphylococcus aureus was confirmed. This has been added in the legend.

-Images with contrast performed : yes but not showed as this case was selected for post-partum changes,

- this is obligatory? This is mandatary in infectious sacroiliitis and, according to your comment 8, this has been added in the text.

- DD enthesitis (see 5). The legend has been corrected and now includes « extensive extra-articular edema and regional muscle edema ». These features are exceptional in enthesitis.

  1. - Can you give the prevalence of variants? The prevalence of the dysmorphic joint and accessory SIJ has been indicated.
  2. - Start with the vessels and their location. Subchondral edema can be mimicked. Elaborate how enthesitis looks like. The text has been corrected accordingly.
  3. –line 218 :add patient’s history (e.g. Radiation therapy). There must be a mistake in the line indicated which is not in chapter 7 in my text. However, radiation therapy as a potential cause of insufficiency fractures has been added in the text and it has been indicated that the patient was osteoporotic in Fig 7.
  4. -For this diagnosis MR with contrast is mandatory- please add this. This has been added.

-Line 233 replace:  dramatic by extensive. This has been corrected.

-Fig 9 (now 10). Please add age and gender of the patient. This has been added.

-There is no fluid, but enhancing synovium seen. The author do not agree with this comment, as in the literature, it has been recommended to avoid the use of SIJ synovitis, as synovium is only present at the perimeter of the lower third of the cartilaginous portion of the joint (Ann Rheum Dis. 2019 Nov;78(11):1550-1558). However, they have added joint space enhancement in the figure of the legend.

-add : no abcès. This has been added.

  1. -The screening for AxSpa is done with only coronal oblique images. So the conclusion would be that in case of doubt axial images can be of help. (Line 259 ) This has been corrected. Please add that the clinical correlation is essential. This has been added.

Reviewer 2 Report

I appreciated very much this paper.

The iconographic part is remarkable.

I suggest You some improvements:

  • Start the paper with a paragraph (add it) with the main MRI features of inflammatory sacroiliitis (a figure should also help); This, should create a frame for your next paper's paragraphs.
  • Please add a very short paragraph in regards to neoplasm differential diagnosis (consider metastasis or multiple myeloma lesions); of course in the majority of cases the d.d. is easy but in small or hill-defined lesions near the joints it could be more difficult.
  • Please discuss if there is any role for Diffusion (DWI)
  • Please add a short paragraph with the optimal MRI sequences (include planes and names of sequences) to assess sacroiliac joint pathology.
  • Re-arrange all the keywords according to Pubmed MESH terms
  • Abstract is missing

Author Response

I appreciated very much this paper. Thank you very much

The iconographic part is remarkable. Thank you very much

I suggest You some improvements:

  • Start the paper with a paragraph (add it) with the main MRI features of inflammatory sacroiliitis (a figure should also help); This, should create a frame for your next paper's paragraphs. The different lesions that can be encountered on MR images have been indicated and a figure has been added. However, this part has not been explained in detail as sacroiilits is not the main topic of this article.
  • Please add a very short paragraph in regards to neoplasm differential diagnosis (consider metastasis or multiple myeloma lesions); of course in the majority of cases the d.d. is easy but in small or hill-defined lesions near the joints it could be more difficult. This paragraph has been added.
  • Please discuss if there is any role for Diffusion (DWI). It has been indicated that « Diffusion-weighted imaging is a sensitive MRI sequence in detection and quantification of active sacroiliitis but is not included in routine protocols » [34]. 
  • Please add a short paragraph with the optimal MRI sequences (include planes and names of sequences) to assess sacroiliac joint pathology. I have only added coronal oblique T1 and fluid sensitive T2-weighted fat-saturated sequences as the choice and possibility of choice of the T2-sequence is mainly influenced by the age and brand of the magnet. It is not so much the type of sequence but the spatial resolution and the quality of the fat signal suppression that are important. This part has not been extended as sacroiliitis is not the main goal of the article.
  • Re-arrange all the keywords according to Pubmed MESH terms. This has been corrected
  • Abstract is missing. Abstract has been added.

Round 2

Reviewer 2 Report

I am satisfied with the revisions performed.

Thanks and Congratulation.